# Docetaxel Loaded in Copaiba Oil-Nanostructured Lipid Carriers as a Promising DDS for Breast Cancer Treatment

**DOI:** 10.3390/molecules27248838

**Published:** 2022-12-13

**Authors:** Fabiola Vieira de Carvalho, Ligia Nunes de Morais Ribeiro, Ludmilla David de Moura, Gustavo Henrique Rodrigues da Silva, Hery Mitsutake, Talita Cesarim Mendonça, Gabriela Geronimo, Marcia Cristina Breitkreitz, Eneida de Paula

**Affiliations:** 1Department of Biochemistry and Tissue Biology, Institute of Biology, University of Campinas-UNICAMP, Campinas 13083-862, Brazil; 2Institute of Biotechnology, Federal University of Uberlandia-UFU, Uberlandia 38405-319, Brazil; 3Department of Analytical Chemistry, Institute of Chemistry, UNICAMP, Campinas 13083-970, Brazil

**Keywords:** breast cancer, nanostructured lipid carriers, docetaxel

## Abstract

Breast cancer is the neoplasia of highest incidence in women worldwide. Docetaxel (DTX), a taxoid used to treat breast cancer, is a BCS-class-IV compound (low oral bioavailability, solubility and intestinal permeability). Nanotechnological strategies can improve chemotherapy effectiveness by promoting sustained release and reducing systemic toxicity. Nanostructured lipid carriers (NLC) encapsulate hydrophobic drugs in their blend-of-lipids matrix, and imperfections prevent drug expulsion during storage. This work describes the preparation, by design of experiments (2^3^ factorial design) of a novel NLC formulation containing copaiba oil (CO) as a functional excipient. The optimized formulation (NLC_DTX_) showed approximately 100% DTX encapsulation efficiency and was characterized by different techniques (DLS, NTA, TEM/FE-SEM, DSC and XRD) and was stable for 12 months of storage, at 25 °C. Incorporation into the NLC prolonged drug release for 54 h, compared to commercial DTX (10 h). In vitro cytotoxicity tests revealed the antiproliferative effect of CO and NLC_DTX_, by reducing the cell viability of breast cancer (4T1/MCF-7) and healthy (NIH-3T3) cells more than commercial DTX. NLC_DTX_ thus emerges as a promising drug delivery system of remarkable anticancer effect, (strengthened by CO) and sustained release that, in clinics, may decrease systemic toxicity at lower DTX doses.

## 1. Introduction

Cancer is the name given to a set of diseases that cause disordered cell growth; the neoplastic cells being able to survive and spread to other tissues and organs [1]. Breast cancer is the most frequent neoplasm in women, worldwide [2,3]. Chemotherapy is the indicated treatment to prevent metastasis, recurrence and to prolong patient survival; the choice of drug or therapeutic regimen is dependent on the staging [4]. Taxanes, such as docetaxel (DTX) and paclitaxel, are chemotherapeutics used in this context [2].

DTX is produced by the esterification of 10-deacetylbaccatin III, a natural compound isolated from *Taxus baccata*. It is a prodrug that, after conversion into the active metabolite, binds to the tubulin β subunit, stabilizing microtubules/preventing their depolarization, interrupting the process of mitosis in the G2 and M phases, causing cell death [5,6]. In addition, DTX prevents tumor growth by other mechanisms: it inhibits angiogenesis, influences the apoptotic pathway [7] and can also promote the expression of the p27 cell cycle inhibitor, and inhibition of the Bcl-2 anti-apoptotic gene expression [8]. These properties confer DTX anticancer activity against a wide range of tumors, including breast, lung, gastric, ovaries and melanoma [9,10,11].

DTX (C_43_H_53_NO_14,_ Figure 1) has a molecular mass of 807.89 g/mol, melting point = 232 °C [12], pKa = 10.97 and l*ogP* = 4.1 [13]; therefore, DTX is barely aqueous-soluble (4.93 µg.mL^−1^) [2,14] and it is classified as a BCS-Class IV compound [15,16]. In the commercial product, (Taxotere^®^) DTX is solubilized by micelles of the non-ionic surfactant Tween 80^®^ and diluted in a 13% ethanol solution prior to intravenous administration [13,17,18,19,20]. In addition, many side effects are associated with DTX therapy (hepatotoxicity, neurotoxicity, hypersensitivity reactions, musculoskeletal toxicity, and neutropenia and fluid retention), leading to the need for strict dosage control [7].

The development of a nanoparticulated drug delivery system (DDS) is a promising alternative to decrease DTX toxicity without affecting its anticancer efficacy [21,22]. Nanostructured lipid carriers (NLC) are DDS that consist of a lipid matrix stabilized by surfactants which are very useful for the encapsulation of hydrophobic drugs. The composition of the lipid matrix defines the subtypes of these lipid-based particles [23]. The first generation, called solid lipid nanoparticles (SLN), were developed in the early 1990s [24,25] as alternative carriers to conventional colloids such as emulsions, liposomes and polymer micelles. Basically, SLN are made up of a solid lipid (SL) matrix, stabilized by surfactant [26,27]. However, the crystallinity of the SL matrix decreases drug upload, and favors drug expulsion during storage [28,29].

The second generation of lipid nanoparticles, called NLC, represents an evolution of SLN, obtained by the addition of at least one liquid lipid (LL) at room temperature [30,31,32] to the structure of nanoparticle. The resulting lipid blend of NLC is not as crystalline as those of SLN, providing higher drug loading and minimizing drug expulsion during storage [4,21,33,34]. NLC can be prepared with excipients derived from natural sources (such as essential oils) which, in addition to exerting a structural function, can bring intrinsic therapeutic properties to the DDS, e.g., antioxidant, anti-inflammatory, antimicrobial and anticancer [35,36,37]. Copaiba balsam oil (CO) and its main components have already been tested as therapeutic excipients of DDS [38,39,40].

CO is extracted from several species of (*Copaifera* L.) trees, largely distributed in Latin America and mainly in Brazil, and it contains many sesquiterpenes and diterpenic acids. One of its main sesquiterpenes, β-caryophyllene (BCP), has anti-inflammatory, antibiotic, antioxidant, anticancer and local anesthetic effects [41,42,43]. In this work, CO was chosen as one LL for NLC preparation, because its anticancer and analgesic properties [44] are desirable to treat tumor progression and pain disorders often associated to breast cancer.

In this work, a new NLC was developed and characterized for the delivery of DTX for the treatment of breast cancer, with therapeutic efficiency and lower toxicity than commercial DTX. The difference of the developed formulation was the use of CO as an excipient with intrinsic therapeutic properties and its optimization through factorial design. The formulation was characterized by different techniques (DLS, NTA, TEM, FE-SEM, DSC and XRD), showed excellent shelf-stability (for 12 months at 25 °C), promoted sustained DTX release and proved to be effective in decreasing the survival of murine breast cancer and non-cancerous cells.

## 2. Results and Discussion

### 2.1. Factorial Design of NLC Formulations

Based on the preliminary lipid-DTX miscibility tests (data not shown), a lipid matrix was chosen for the nanoparticles, composed of myristyl myristate (MM), Miglyol 812^®^ (MIG) and CO. Pluronic F-68 (P68) was the chosen surfactant. The structures of the major NLC excipients and DTX are given in Figure 1. The variables LL, P68 and DTX were subjected to a 2^3^ factorial design, to define the optimal excipient concentrations. Table 1 shows the results obtained with the 11 tested compositions and the measured properties of interest (responses): size, polydispersity index (PDI) and zeta potential (ZP). Surface graphs were provided for each statistically significant response, as shown in Figure 2A–C, including a desirability graph (Figure 2D). Finally, the variables with higher influences on the studied properties of interest are given in Appendix A.

Particle size is one of the main parameters employed to passively target therapeutic agents to tumors [45]. The submicron size of the nanocarriers used in DDS favors their targeting to the tumor tissue, and the vasculature presents pathophysiological and anatomical imperfections (fenestrated capillaries), allowing the passive entry of nanoparticles, due to the so-called enhanced permeability and retention (EPR) effect [45,46]. The particle sizes ranged from to 144.1 and 196.8 nm (Table 1). As expected, P68 exerted a negative effect on particle size (Figure 2A, Appendix A), since surfactants are known to reduce the surface tension and stabilize particles in suspension. In addition, the hydrophilic part of this block-copolymer surfactant produces a steric hindrance that prevents particle aggregation [47,48]. The variation of the other factors had no influence on the size response.

The polydispersity indexes varied between 0.125 and 0.175 (Table 1). PDI indicates the degree of homogeneity of particle sizes, with lower values (<0.2) being desirable and indicative of monodisperse systems [49,50]. DTX, P68 and the interaction between DTX with LL (AC) were responsible for increasing PDI (Figure 2B,C and Appendix A). ZP is a parameter related to the surface charge of particles that is correlated with the stability of colloidal systems [24]. In all tested formulations, ZP values were negative and higher than |20| mV (Table 1), ensuring the stability of the NLC [48,51]. 

From the predetermined criteria (lower sizes and PDI and higher ZP values, in modulus—see Section 3.3. of methods) a desirability graph was generated (Figure 2D), revealing the desirability parameter that directly (the closer to “1”, the better) indicates the formulation that meets the stablished criteria. For the NLC_DTX_, desirability (0.593) revealed that the conditions near the central point better met the desired criteria; therefore, the formulation containing 80% LL, 10% P68, 0.5% of DTX was chosen. The optimized formulation, as well as a control (prepared without DTX), were prepared again and their physicochemical stability was monitored for particle size, PDI ZP, number of particles.mL^−1^ and encapsulation efficiency (% EE) values for 12 months, at 25 °C.

### 2.2. Characterization of the Optimized NLC Formulation

#### 2.2.1. Size, PDI, ZP, % EE and Drug Loading (% DL)

The optimized formulation (NLC_DTX_) and its control, prepared without DTX (NLC_CTL_) were characterized and their physicochemical properties are given in Table 2.

The size of the nanoparticles was slightly higher in the presence of DTX, but the low PDI values (<0.2) of both samples reveal an uniform size distribution [49,50]. In addition, the ZP values were far from neutral, as desirable for pharmaceutical formulations of long-term stability [24]. The optimized formulation showed superior upload capacity (% EE ~100% and DL = 1.5%) compared to similar lipid-based nanoparticulated DDS as already reported in the literature: 94% EE and % DL = 1.3 [52,53].

#### 2.2.2. Concentration of Nanoparticles

The number of particles per milliliter (4.2 × 10^13^ for NLC_DTX_ and 4.1 × 10^13^ for NLC_CTL_) were determined by Nanotracking analysis (NTA) and were in the same order of magnitude registered for similar NLC systems in the literature [54,55,56]. NTA also provided information of the particle size (ca. 185 nm) and distribution (Span values), as shown in Appendix A. The observed Span values smaller than 1 confirmed the homogeneous size distribution of the optimized formulation [55], in agreement with the DLS results.

#### 2.2.3. Transmission and Scanning Electron Microscopy

TEM and FE-SEM images provide information on morphology of the NLC. As expected for these kinds of lipid carriers, spherical particles with smooth surfaces prevailed in both formulations: with and without DTX (Figure 3). It is also worth mentioning that the addition of DTX to the system did not change the morphology of the nanoparticles (Figure 3A–C vs. Figure 3D–F). The sizes of the nanoparticles in the micrographs (~200 nm as measured using the ImageJ software (version 1.53k, NIH, Bethesda, MD, USA) show good agreement with those reported by DLS and NTA (Table 2 and Appendix A, respectively).

#### 2.2.4. Differential Scanning Calorimetry (DSC) an X-ray Diffraction (XRD) Analyses

DSC and XRD techniques provide information about the crystallinity of the lipid NLC matrix. The measurement of the transition energy of the major lipid excipient DSC thermograms reveal information on the lipid core packing that may determine drug loading by the nanoparticles [57].

Figure 4A shows the thermograms of the major excipients (MM, P68) used in the composition of the NLC, and of DTX. In accordance with the literature, the melting point of MM was registered at 45.46 °C [58], P68 at 54.69 °C [59,60], while DTX showed an endothermic peak around 166.64 °C [2,61,62]. Thermograms were also run for the physical mixtures (PM) of excipients without DTX (PM_CTL_), with DTX (PM_DTX_) and for NLC_CTL_ and NLC_DTX_.

The melting point of PM_CTL_ (47.49 °C) mainly reflected the transition of the major (MM) excipient while that of PM_DTX_ was measured at a slightly higher temperature (51.11 °C), probably reflecting the presence of 0.5% DTX. Finally, the endothermic peaks determined in the optimized formulations (NLC_CTL_ = 51.53 °C and NLC_DTX_ = 50.21 °C) were broadened and between those of the NLC major excipients, as evidence of their (MM, P68) molecular rearrangement inside the lipid nanoparticles.

According to the X-ray diffractograms in Figure 4B, the major NLC excipients show two characteristic peaks of high intensity at 21° and 24° (MM) [58] and at 19° and 23° (P68) [63]. For pure DTX, significant diffraction peaks were detected at 2θ scattering angles between 8° to 20°, in agreement with previous published data [52,64,65,66]. The narrow peak of MM (at 21°) decreased in intensity in the physical mixture (PM_CTL_ and PM_DTX_), and almost disappeared when the nanoparticles were formed (NLC_CTL_ and NLC_DTX_). Such lower peak intensities, compared to PM or pure excipients, indicate loss of the crystalline structure of the lipid core. Finally, the diffraction patterns were similar for particle samples containing DTX (NLC_DTX_) or not (NLC_CTL_), confirming that the drug did not change the general organization of these nanoparticles, as also indicated by TEM/FE-SEM data (Figure 3).

### 2.3. Stability Studies

Figure 5 shows the changes in physicochemical properties of the optimized NLC and its control, during 12 months of storage at room temperature.

After 12 months of storage no significant changes were observed, compared to the initial values (*p* > 0.05), for any of the parameters (size, PDI, ZP, nanoparticles concentration and % EE). In addition, no macroscopic signs of instability (drug aggregation/precipitation) were observed, indicating excellent shelf-stability for the NLC. It is worth noting the importance of measuring both the average size and number of particles in suspension over the time, since these parameters would be inversely correlated (the higher the size, the smaller the number of particles) in case of particle aggregation [50]. Therefore, our results attest that no fusion occurred during storage. The stability in terms of % EE also confirmed that DTX was not expelled from the NLC, even after one year of storage at 25 °C.

### 2.4. In Vitro Tests

#### 2.4.1. Release Kinetics

Figure 6 shows the results of the in vitro release kinetics of DTX, free or encapsulated, in the optimized NLC formulation.

Under the experimental conditions, a more pronounced release was observed in the first 10 h, with approximately 100% and 45% of DTX released from commercial (free DTX) and NLC_DTX_ samples, respectively (Figure 6). After this time and up to the end of the experiment, NLC_DTX_ exhibited a sustained release, so that complete release was achieved only after 54 h. Interestingly, no initial burst phase was observed for NLC_DTX_, probably because the antineoplastic was completely encapsulated (EE > 99%) by the nanoparticles. Similar results were described by Liu et al. for the encapsulation of DTX in NLC composed of stearic acid, glyceryl monostearate, soya lecithin, oleic acid and P68 [9] indicating that the slower release reflects the higher affinity of the antineoplastic for these lipid-based nanoparticles.

The prolonged release of antineoplastic drug from the NLC may be correlated with a decrease of the in vivo systemic toxicity, after administration [23]. Different mathematical models have been used to describe the release profiles of lipid, polymer and lipid-polymer DDS loading different active-molecules [53,60,64,67,68]. To better understand the mechanisms ruling DTX liberation, the curves in Figure 6 were fitted by several kinetic models, and the best fit for free DTX and NLC_DTX_ kinetics was found with the Weibull (R^2^ = 0.9522) and zero order (R^2^ = 0.9486) equations, respectively (Appendix A). The Weibull model (a normal distribution) defined the release profile of DTX from the Tween 80 micelles, present in the commercial DTX formulation. As for the optimized NLC, the zero order model revealed a regimen of continuous drug release [69], also found by Gao et al. for DTX encapsulated in albumin-lipid nanoparticles [6]. This continuous release of DTX is consistent with the outstanding encapsulation efficiency (>99%) of the nanoparticles. Indeed, taking into account the concentration (0.5%) and molar mass (807.88 g/mol) of docetaxel and excipients, plus the number of particles in suspension (4.2.10^13^/mL—Appendix A), we estimated the number of DTX molecules (8.8.10^4^) per nanoparticle [60]. Knowing that DTX represents no more than 1 mol % of the molecules inside each NLC (excipients molar ratio: solid lipid = 25%, liquid lipid = 71% and P68 = 3%), it seems that the rate limiting step of the zero order release (NLC_DTX_ curve) is the slow diffusion of DTX from the lipid NLC matrix to the aqueous milieu [70].

#### 2.4.2. Cell Viability Tests

The effect of DTX, CO, NLC_CTL_ and NLC_DTX_ were tested over non-cancerous (NIH-3T3 murine fibroblasts), and breast cancer (4T1 and MCF-7, murine and human, respectively) cell lineages for 24 and 72 h. In order to clarify the effect of CO, another NLC formulation was prepared with only the synthetic liquid lipid (MIG) and without CO (Synth-NLC, Synth-NLC_DTX_) and its effect was tested.

Figure 7 shows the results of the treatment of 4T1, MCF-7 and NIH-3T3 cells for 24 h with the samples: CO, NLC_CTL_, Synth-NLC_CTL_, DTX, NLC_DTX_ and Synth-NLC_DTX_. In the samples without DTX (Figure 7A,C,E) the X axis was expressed either in CO concentration (% *w*/*w*) or by the number of particles.mL*^−^*^1^ (NLC_CTL_, Synth-NLC_CTL_).

As expected, the control formulations-NLC_CTL_ and Synth-NLC_CTL-_ show very low cytotoxicity against the three cell lines tested, with significant decreases (*p* < 0.05) in cell viability only at high nanoparticle concentrations (>10^11^ particles.mL^−1^). This result also corroborates the literature that reports that the number of nanoparticles is the determinant in the cellular toxicity response [47,71].

The docetaxel-containing formulations-DTX, NLC_DTX_ and Synth-NLC_DTX-_ caused a significant reduction (*p* < 0.05) in cell viability, both in tumor (4T1 and MCF-7, Figure 7B,D) and in non-tumor (NIH-3T3, Figure 7F) cells, and is very evident at the higher concentrations tested (0.3 mM and 3 mM). The cytotoxicity profiles were dose-dependent and varied according to the susceptibility of each cell line. At the highest concentration tested (3 mM) cell viability was reduced to 26.5 ± 1.3% (DTX), 14.0 ± 0.7% (NLC_DTX_) and 37.7 ± 8.1% (Synth-NLC_DTX_) in 4T1 cells; to 27.7 ± 0.8% (DTX), 34.7 ± 2.6% (NLC_DTX_) and 52.4 ± 3.2% (Synth-NLC_DTX_) in MCF-7; and to 26.5 ± 2.9% (DTX), 32.7 ± 3.7% (NLC_DTX_) and 45.6 ± 9.2% (Synth-NLC_DTX_) in NIH-3T3 cells. Taking the results obtained with human (MCF-7) breast cancer cells, even at 0.3 mM (clinical dose concentration), the effect of NLC_DTX_ over free DTX was not as pronounced as those observed with 4T1 and NIH-3T3 cells and surpassed that of Synth-NLC_DTX_. The cell viability of the 4T1 tumor line was the most affected by the developed NLC_DTX_ followed by NIH-3T3.

Considering the good response of NLC_DTX_ over 4T1 breast cancer cells, the effect was also evaluated after 72 h of treatment (Appendix A). The 72 h results show that 4T1 cell viability was decreased by 50% even at lower concentrations (0.0003 mM) of DTX, NLC_DTX_ and Synth-NLC_DTX_. As expected, the effect of control formulations (Appendix A) was not pronounced over time, although the presence of copaiba oil (in CO and NLC_DTX_ samples) was evident, mainly at the concentration of 0.07% (*w*/*w*) and above (as discussed below). The time-dependent cytotoxicity induced by NLC_DTX_ over 4T1 cells is clearly demonstrated in Appendix A.

Since NLC_DTX_ was found more cytotoxic than Synth-NLC_DTX_ formulations against the three cell lines tested (Figure 7B,D,F), we decided to check the effect of CO (Figure 7A,C,E). Our tests revealed that CO, from concentrations of 0.07% *w*/*w* and higher, promoted a significant (*p* < 0.05) reduction in cell viability, especially for tumor cells: 54.1 ± 4.1% (4T1, Figure 7A) and 30.8 ± 2.2% (MCF-7, Figure 7C) against 63.3 ± 6.7 of NIH-3T3 fibroblasts (Figure 7E). These results, attributed to the intrinsic cytotoxic activity of copaiba oil, were also observed after treatment with NLC_CTL_, but not with Synth-NLC_CTL_. As previously discussed, CO has antiproliferative properties [41,42,43]. This natural excipient of NLC_CTL_ (and NLC_DTX_) has a wide pharmacological application in the treatment of neoplastic melanoma [72], skin carcinoma [73] and invasive micropapillary carcinoma [74].

Indeed, the half maximal inhibitory concentration (IC_50_) [75] determined after 24 h of treatment in CO, NLC_CTL_ and NLC_DTX_ samples confirmed the antiproliferative effect of copaiba oil, as shown in Table 3.

The cytotoxic effect of CO over the three cell lineages was confirmed. Encapsulation into NLC heightened the CO antiproliferative effect against tumor and non-tumor cells. Moreover, the CO effect seems to be synergic to that of DTX, considering the lower IC_50_ values determined with NLC_DTX_ regarding those with NLC_CTL_, in each cell lineage tested.

The IC_50_ values determined after 24 h of treatment, concerning the effect of docetaxel (DTX, NLC_DTX_ and Synth-NLC_DTX_) are given in Appendix A. As observed by other authors, the IC_50_ values for MCF-7 cells were significantly higher than those for 4T1 and NIH-3T3 cells, revealing the resistance of this human breast cancer lineage [2,9]. Unfortunately, the standard deviation of the determined IC_50_ values were too high and compromised the assessment of CO and DTX effects into the NLC_DTX_. The discussion of the NLC_DTX_ effect over the tumor cells was then restricted to the results in Figure 7 and Appendix A.

## 3. Materials and Methods

### 3.1. Materials

Docetaxel powder (DTX) was a gift from Cristália Prod. Quim. Farm. Ltda (Itapira, Brazil). DTX in solution—a generic form of Taxotere^®^—was a gift from Blau Farmacêutica S.A. (Cotia, Brazil). Pluronic F-68 (P68), Dulbecco’s Modified Eagle Medium (DMEM), fetal bovine serum (FBS), 3-(4,5-dimethylthiazol-2-yl)-2,5-diphenyltetrazolium bromide (MTT), penicillin, streptomycin sulphate and trypsin were supplied by Sigma Chem. Co. (St. Louis, MO, USA). Murine fibroblasts (NIH-3T3) and breast cancer cell lines (4T1 and MCF-7) were purchased from American type culture collection (ATCC, Manassas, VA, USA). Copaiba oil (CO) and Miglyol 812^®^ (MIG) were purchased from Engenharia das Essências (São Paulo, Brazil). Myristyl myristate (MM) was purchased from Croda (Campinas, Brazil). Dimethyl sulfoxide (DMSO) was purchased from Laborclin (Pinhais, Brazil). HPLC-grade methanol was purchased from J.T. Baker (Allentown, PA, USA). Deionized water (18 MΩ) was obtained with an Elga USF Maxima ultra-pure water purifier.

### 3.2. NLC Preparation

In the preparation of NLC, an emulsification-ultrasonication method was used [21,76]. First, the oil phase—composed of solid and liquid lipids and DTX (powder)—was heated to 50 °C in a water bath while an aqueous phase (P68 solution) was heated to the same temperature. Both phases were mixed under high-speed agitation (11,000 rpm) for 3 min in an Ultra-Turrax homogenizer (IKA Werke, Staufen, Germany). The mixture was then sonicated for 16 min using a Vibracell tip sonicator at 20 kHz/150 W (Sonics & Mat. Inc., Danbury, CT, USA), in alternating 30 s on/off cycles. Finally, the resulting emulsion was rapidly cooled in an ice bath to 25 °C [10,60,77]. For the preparation of NLC control, the same procedure was adopted, except for the addition of DTX to the oil phase. At the end of preparation, all samples were stored at 25 °C for future testing.

MM was the SL of all formulations. For the LL, a mixture of MIG:CO (67:13% *w*/*w*) was used. Only for the sake of comparison in cytotoxicity tests, an additional formulation was prepared without CO (Synth-NLC) and only MIG, a synthetic triglyceride of caprylic and capric fatty acids, as the LL.

### 3.3. Factorial Design

A 2^3^ factorial design with central points in triplicate was performed using Design Expert software (version 13, Stat-Ease Inc., USA). Analysis of Variance (ANOVA) was applied to verify the regression significance and to identify the most important experimental variables (*p*-value < 0.05). The independent variables and their levels are listed in Table 4, as well as the optimization criteria applied to the analyzed responses: nanoparticle size, polydispersity index (PDI) and zeta potential (ZP). The total amount of lipids was kept at 23% *w*/*w*. The 11 formulations are described in Table 1.

### 3.4. NLC Characterization

#### 3.4.1. Size, PDI, Zeta Potential and Concentration of Nanoparticles

The hydrodynamic diameter (size) and polydispersity index (PDI) were determined by dynamic light scattering (DLS) while the zeta potential (ZP) was measured by electrophoretic mobility in a Nano ZS90 analyzer (Malvern Instruments, UK), at 25 °C. The samples were diluted (1000×) in deionized water. The concentration of nanoparticles in the formulations was determined by Nanotracking analysis (NTA) in a NS300 (NanoSight, Amesbury, UK). The samples were diluted (50,000×) in deionized water and injected into the sample chamber with syringes. All measurements were performed at 25 °C (n = 3). Particle size distribution was determined by the Span index, and calculated according to Equation (1) [78]:(1)$$SPAN=\frac{D90\%-D10\%}{D50\%}$$

#### 3.4.2. DTX Quantification, Encapsulation Efficiency and Drug Loading Determination

Docetaxel was quantified using high-pressure liquid chromatography (HPLC) in a Waters Breeze 2 (Waters Technologies, Milford, MA, USA) and a C18 Gemini, 5 µM, 150 × 4.60 mm column, at 35 °C. The mobile phase was a mixture of methanol:water 70:30 (*v*/*v*). The flow rate, injection volume and wavelength detection were set at 1 mL.min^−1^, 20 µL and 270 nm, respectively [79]. The encapsulation efficiency (% EE) of DTX by the nanoparticles was determined by the ultrafiltration–centrifugation method, using cellulose filters (10 kDa, Millipore). Briefly, the total amount (100%) of DTX in the NLC was determined (DTX_total_) by diluting the samples in the mobile phase (n = 3). The amount of DTX in the filtrate (DTX_free_) was quantified by HPLC and the percentage of encapsulated DTX was calculated according to equation 2. Drug loading, the amount of loaded DTX in relation to the total weight of the nanoparticles, was also calculated according to Equation (3) [10]:(2)$$\%\;\mathrm{EE}=\frac{\mathrm{DTXtotal}-\mathrm{DTX}\;\mathrm{free}}{\mathrm{DTX}\;\mathrm{total}}\times 100$$
(3)$$\%\;Drug\;Loading=\frac{weight\;of\;encapsulated\;DTX\;}{weight\;of\;nanoparticles}\times 100$$

#### 3.4.3. Transmission (TEM) and Scanning (FE-SEM) Electron Microscopy

The morphology of the optimized formulation (NLC_DTX_) and its control (NLC_CTL_) was elucidated using TEM and FE-SEM.

Uranyl acetate (2%) was added to the appropriately diluted samples to provide contrast for TEM. Aliquots (diluted 33× in deionized water) were deposited onto copper grids coated with a carbon film and dried at ambient temperature. After drying, micrographs of the samples were obtained using a Tecnai G2 Spirit BioTWIN (FEI Company, Hillsboro, OR, USA) microscope operated at 60 kV and the images were edited with ImageJ software v.1.53k (https://imagej.nih.gov/ij/, accessed on 23 March 2022).

For the FE-SEM sample preparation, a drop of each sample was adhered to a glass coverslip previously nailed to an aluminum sample holder (stub). After complete evaporation of the solvent, the stubs were subjected to the sputtering process for 120 s at 30 kV and stored in a desiccator until analysis. The samples were visualized in a Zeiss EVO MA10 scanning electron microscope with secondary and backscattered electron detectors, operating in a high vacuum under a voltage of 20 kV.

#### 3.4.4. Differential Scanning Calorimetry (DSC) and X-ray Diffraction (XRD) Analysis

Thermograms of the optimized formulations (NLC_DTX_ and NLC_CTL)_, their physical mixture and excipients were obtained with the DSC 2910 calorimeter, in a standard sample holder and analyzed with the Thermal Solutions v.1.25 program. Samples of formulations were lyophilized prior to analysis. The heating rate used was 10 °C min^−1^, in the range of 20 to 180 °C.

X-ray diffraction (XRD) measurements were carried out from 2θ = 5–50°, at a rate of 2° min^−1^, in a XRD7000 (Shimadzu, Japan) diffractometer with a Cu-Kα source for identification of the crystalline structures in the lipid matrix of NLC (the samples were lyophilized before the analysis).

### 3.5. Physicochemical Stability Study

The physicochemical stability at 25 °C and 60% relative humidity (RH) of the optimized formulation and its control was evaluated at predetermined time intervals, for 12 months. The analyzed parameters were nanoparticle size and concentration, PDI, ZP and % EE as well as the visual inspection of formulation appearance. Analysis of variance (ANOVA, *p* < 0.05) was used to compare the initial and subsequent values.

### 3.6. In Vitro Tests

#### 3.6.1. Release Kinetics Tests

For the in vitro release kinetics assays, 15 mL Franz-type vertical diffusion cells (0.6 cm^2^ permeation area) were used. In the donor compartment, the formulation (NLC_DTX_) or a commercial DTX (Docetaxel 20 mg. mL^−1^, Blau Farm. S.A.) was added. The receptor compartment was filled with a mixture of 5 mM phosphate-buffered saline solution (PBS) pH 7.4, ethanol and Tween 80 (80:15:5 % *v*/*v*) [52,80,81]. In this system, a polycarbonate membrane (47 mm diameter, 100 nm molecular exclusion pore, Nuclepore Track-Etch Membrane, Whatman^®^) was used as a barrier to separate the sample (0.2 mL in the donor compartment) from the recipient compartment. The solution in the receiving compartment was kept under gentle agitation (300 rpm) at 37 °C throughout the experiment. At determined intervals (0 to 54 h, until complete release), 0.2 mL of sample was withdrawn from the receiving compartment (and the volume was replaced with the PBS: ethanol: Tween 80 solution) for DTX quantification by HPLC. KinetDS 3.0 software was used to analyze the release curves [82]. A few mathematical models were tested (zero order, first order, Korsmeyer-Peppas and Weibull) and those that best described the release curves were selected accordingly to the coefficient of determination (R^2^). The best models (zero order and Weibull) [59,83] are described by Equations (4) and (5), respectively:(4)$$Q\;=\;k\times t\;+Q_{0}$$
where Q is the amount of drug released over time (t), Q_0_ is the initial value of Q and k is the release constant.
(5)$$Q=1-\exp\left[\frac{-t^{n}}{a}\right]\;$$
where a is a scale parameter that describes the time dependence, and n the parameter that characterizes the release curve.

#### 3.6.2. Cell Viability Tests

Balb/c 3T3 murine fibroblasts (NIH-3T3), murine breast cancer (4T1) and human breast cancer (MCF-7) cell lines were used, and viability was measured by the MTT test. Cells were cultured in DMEM medium supplemented with 10% fetal bovine serum and 1% antibiotic (penicillin and streptomycin). The cells were plated in 75 cm^2^ bottles an incubated (Shel Lab—CO_2_ incubator, USA) at 37 °C under a humidified 5% CO_2_ atmosphere for 48 h, until semi-confluence. Then, the 4 × 10^4^ cells/well (NIH-3T3 and 4T1) or 5 × 10^4^ cells/well (MCF-7) were incubated in 96-well plates for 24 h at 37 °C under a humidified atmosphere and 5% CO_2_. After this period, pure medium was replaced by medium containing the respective treatment groups (Table 5), in serial dilutions (9) to different concentrations (from 0.03 nM to 3 mM), and the treatments were performed for 24 and 72 h. After treatment the medium was removed, the plates washed with sterile PBS, and 150 μL of MTT (0.5 mg/mL in culture medium) was added to each well. After incubation for 3 h at 37 °C, the MTT solution was removed, and the formazan crystals formed were solubilized in 150 μL DMSO. The plates were shaken for 10 min, and the absorbance of each well was read at 570 nm. Values were expressed as percent MTT reduction, in comparison to control (untreated cells). The analyses were performed in triplicate using GraphPad Prism 8.0 software, and the results expressed as mean ± SD. The statistical analysis of the results was performed by two-way ANOVA, followed by Tukey’s test (*p* < 0.05).

## 4. Conclusions

In the search for a safer and more effective treatment for breast cancer, a novel nanostructured lipid carrier loading docetaxel (NLC_DTX_) was developed through the design of experiments approach. An optimized NLC_DTX_ formulation with excellent DTX encapsulation efficiency and long shelf-stability was obtained. The formulation, containing copaiba oil as a functional (LL) excipient, promoted sustained drug release and significantly reduced cell viability of tumor (4T1***/***MCF-7) and non-tumor (NIH-3T3) cells, in a dose-dependent manner and to higher levels than commercial DTX (4T1).

The NLC_DTX_ effect on cell viability was always higher than that of NLC_CTL_ (prepared without docetaxel), indicating a synergism between CO + DTX (Table 3). CO alone also promoted a significant (*p* < 0.05) reduction in cell viability especially for tumor cells, from the concentration of 0.07% *w*/*w* and higher.

The NLC_DTX_ effect on human adenocarcinoma (MCF-7 cells) was not pronounced, probably because of the higher resistance of this lineage [2,9] to any of the formulations, in the concentration range tested. On the other hand, at the higher concentrations tested (0.3 and 3 mM), NLC_DTX_ was found to be more toxic to murine adenocarcinoma (4T1 cells) than to non-cancerous (3T3) cells, as shown in Figure 7B,F. This effect could be interesting in clinics, if lower doses of NLC_DTX_ (e.g., at 0.3 mM produced more cell death in 4T1 tumor cells than that evinced by DTX at 3 mM), were as effective as the commercial product, lowering DTX systemic toxicity.

The antitumor effects of the NLC_DTX_ formulation also open new perspectives to better understand the mechanisms of cell death promoted by the association of copaiba oil with DTX. In the next steps of this study, we intend to conduct in vivo efficacy (anticancer and antinociceptive) assays, to confirm the potential of NLC_DTX_ as a less toxic and more effective DDS for the treatment of breast cancer.

## Figures and Tables

**Figure 1 molecules-27-08838-f001:**
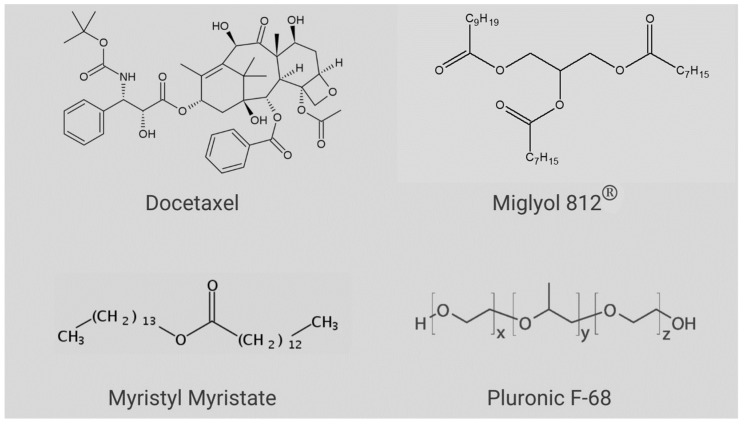
Chemical structures of the compounds used in NLC preparation and docetaxel.

**Figure 2 molecules-27-08838-f002:**
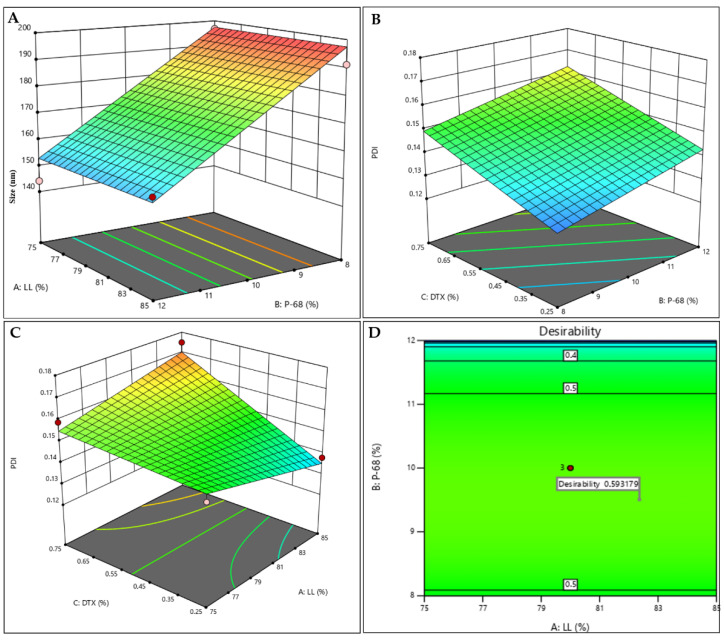
Factorial design results for the NLC_DTX_ system: response surfaces for size (**A**), PDI (**B**,**C**), and desirability graph (**D**).

**Figure 3 molecules-27-08838-f003:**
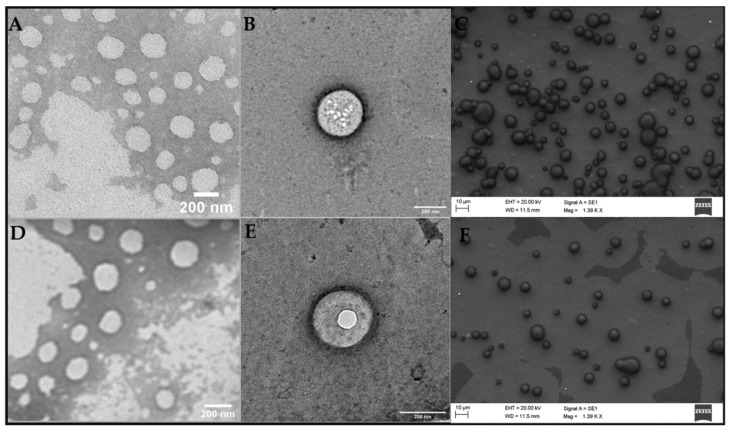
TEM (**A**,**B**,**D**,**E**) and FE-SEM (**C**,**F**) micrographs of the optimized formulation: NLC_DTX_ (**A**–**C**) its control, and NLC_CTL_ (**D**–**F**). Magnitudes: 30,000× (**A**,**D**), 49,000× (**B**,**E**) and 1390× (**C**,**F**).

**Figure 4 molecules-27-08838-f004:**
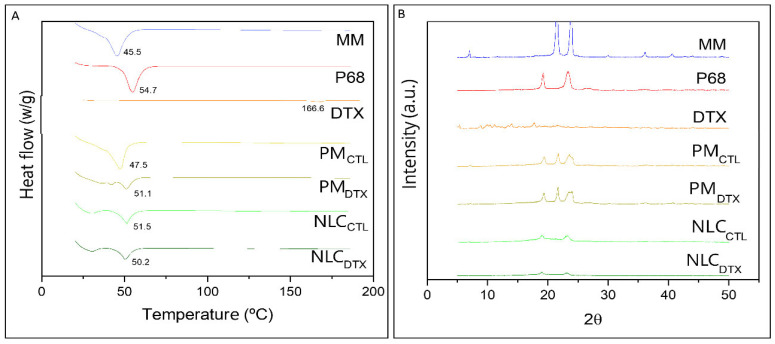
(**A**) DSC thermograms of pure myristyl myristate (MM), Pluronic F-68 (P68), Docetaxel (DTX), physical mixture of excipients without DTX (PM_CTL_), with DTX (PM_DTX_), NLC without (NLC_CTL_) and with DTX (NLC_DTX_) determined at a heating rate of 10 °C min^−1^; (**B**) X-ray diffractograms of: MM, P68, DTX, PM_CTL_, PM_DTX_, NLC_CTL_ and NLC_DTX_ obtained using a Cu-Kα source, at 2° min^−1^.

**Figure 5 molecules-27-08838-f005:**
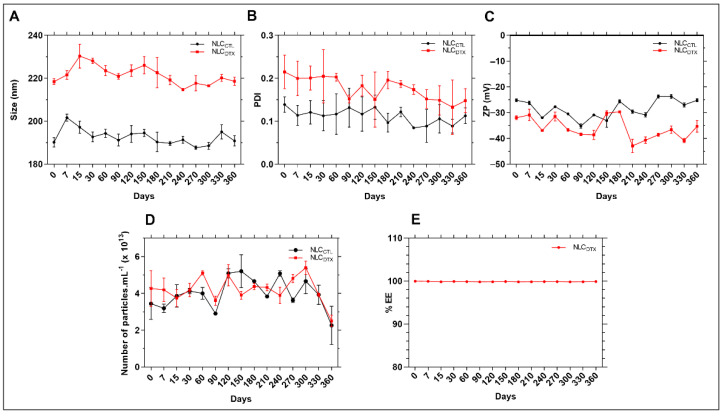
Stability data of NLC_CTL_ and NLC_DTX_ formulations in relation to (**A**) size, (**B**) PDI, (**C**) ZP, (**D)** number of particles/mL and (**E**) DTX encapsulation efficiency (% EE), during 12 months of storage at 25 °C. Two-way ANOVA plus Tukey-Kramer post hoc tests revealed no statistical differences (*p* > 0.05) after one year, in comparison to the values determined for freshly prepared samples, for in any of the analysed (size, PDI, ZP, number of particles or % EE) parameters.

**Figure 6 molecules-27-08838-f006:**
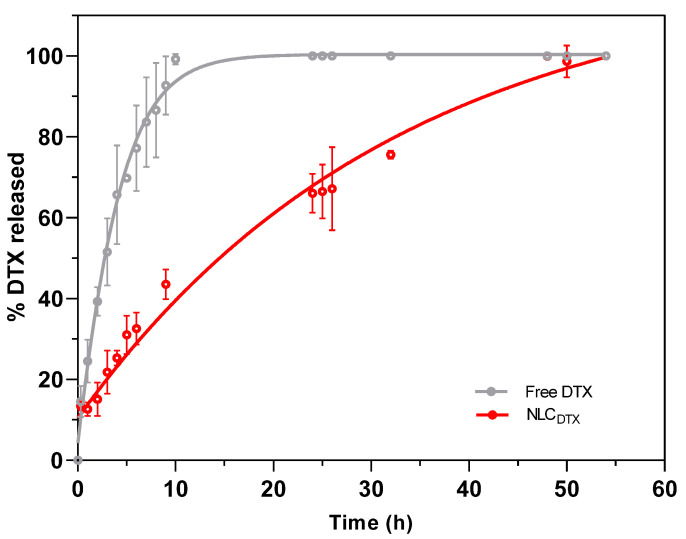
In vitro release kinetics of free or encapsulated DTX (NLC_DTX_) determined at 37 °C, with a receptor medium composed of PBS: ethanol: Tween 80 (80:15:5%, *v*/*v*), n = 3.

**Figure 7 molecules-27-08838-f007:**
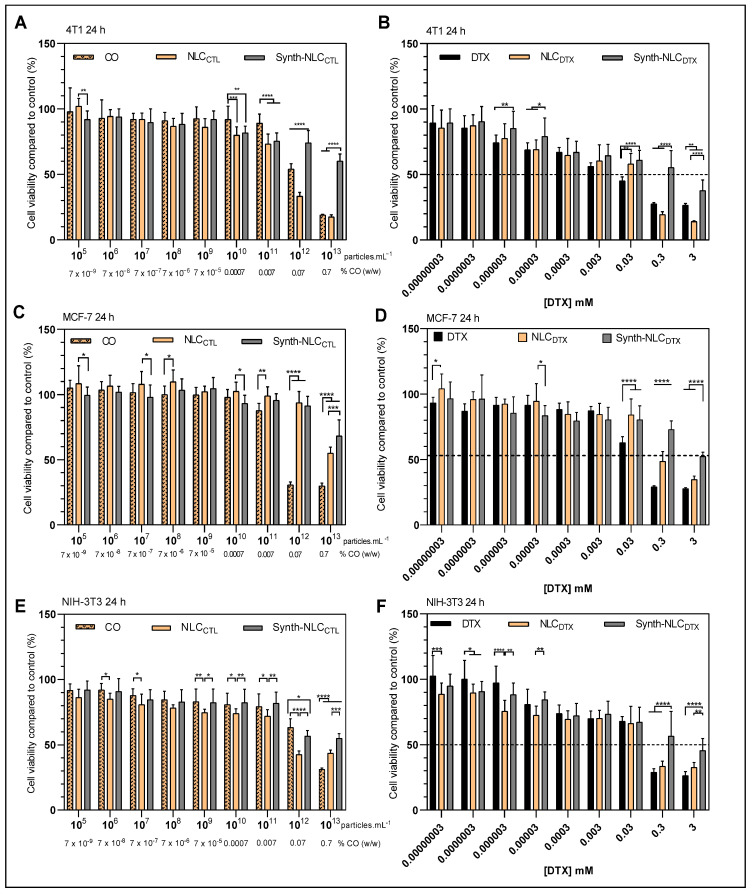
Cell viability (MTT assay) in 4T1 (**A**,**B**), MCF-7 (**C**,**D**) and NIH-3T3 (**E**,**F**) cells treated for 24 h with control samples: CO, NLC_CTL_ and synth-NLC_CTL_, (**A**,**C**,**E**) and docetaxel-containing samples: free (DTX) or encapsulated in the optimized (NLC_DTX_) or synthetic (Synth-NLC_DTX_) nanoparticles (**B**,**D**,**F**). Results expressed as the mean ± SD (n = 3). Statistical analysis by two-way ANOVA plus Tukey–Kramer post hoc. * *p* < 0.05; ** *p* < 0.01; *** *p* < 0.001, **** *p* < 0.0001.

**Table 1 molecules-27-08838-t001:** Results obtained from the 2^3^ factorial design of NLC_DTX_, showing the independent (LL, P68 and DTX) and dependent variables (responses): size, PDI and ZP. The independent variables are also identified by capital letters: A = LL, B = P68, C = DTX.

	Variables			Responses		
Formulation	A: LL (% *w*/*w*)	B: P68 (% *w*/*w*)	C: DTX (% *w*/*w*)	Size (nm)	PDI	ZP (mV)
1	75	8	0.25	196.8	0.143	−26.5
2	85	8	0.25	191.2	0.125	−24.0
3	75	12	0.25	144.1	0.147	−26.6
4	85	12	0.25	154.7	0.138	−25.7
5	75	8	0.75	187.3	0.140	−27.1
6	85	8	0.75	182.2	0.158	−25.1
7	75	12	0.75	163.3	0.159	−23.9
8	85	12	0.75	155.9	0.175	−24.6
9	80	10	0.5	179.0	0.137	−24.8
10	80	10	0.5	185.0	0.146	−27.5
11	80	10	0.5	189.5	0.139	−24.8

**Table 2 molecules-27-08838-t002:** Physicochemical properties (size, PDI, PZ, % EE and % DL) of the NLC formulation selected by factorial design for DTX delivery, and its control.

Formulation	Size (nm)	PDI	ZP (mV)	% EE	% DL
**NLC_DTX_**	221.5 ± 2.5	0.18 ± 0.03	−36.0 ± 1.2	99.87 ± 0.01	1.49
**NLC_CTL_**	192.6 ± 2.3	0.11 ± 0.03	−28.4 ± 0.7	-	-

**Table 3 molecules-27-08838-t003:** IC_50_ values determined for CO against 4T1, MCF-7, and NIH-3T3 cells, after 24 h of treatment and measured by the MTT assay. Analysis made with the GraphPad Prisma 8 software of the values taken from the curves in Figure 7.

	Cancer Cell Lines		Non-Cancerous Cell Line
	4T1 (Murine)	MCF-7 (Human)	NIH-3T3 (Murine)
Formulation	IC_50_ (% *w*/*w*)	IC_50_ (% *w*/*w*)	IC_50_ (% *w*/*w*)
**CO**	0.053 ± 0.020	0.012 ± 0.004	0.044 ± 0.028
**NLC_CTL_**	0.006 ± 0.004	0.007 ± 0.004	0.004 ± 0.003
**NLC_DTX_**	0.002 ± 0.001	0.006 ± 0.002	0.001 ± 0.001

**Table 4 molecules-27-08838-t004:** Experimental variables, levels and responses analyzed in the 2^3^ factorial design.

**Variables**	**Symbols**	**Low Level**	**High Level**
**LL (% *w*/*w*)**	A	75	85
**P68 (% *w*/*w*)**	B	8	12
**DTX (% *w*/*w*)**	C	0.25	0.75
**Responses**	**Optimization**		
**Size (nm)**	Lowest		
**PDI**	Lowest		
**ZP (|mV|)**	>20		

**Table 5 molecules-27-08838-t005:** In vitro treatment groups. The commercial DTX (containing 20 mg mL*^−^*^1^) was a gift from Blau Farm. S.A.

Treatments	Abbreviations
Commercial DTX	DTX
Copaiba oil (balsam)	CO
NLC with CO and without DTX	NLC_CTL_
NLC with CO and with DTX	NLC_DTX_
NLC without CO and without DTX	Synth-NLC_CTL_
NLC without CO and with DTX	Synth-NLC_DTX_

## Data Availability

The data presented in this study are available on request from the corresponding author.

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
