# Peer review of "Docetaxel Loaded in Copaiba Oil-Nanostructured Lipid Carriers as a Promising DDS for Breast Cancer Treatment"

_molecules, 2022, doi:10.3390/molecules27248838_

Round 1

Reviewer 1 Report

The manuscript reported by Fabíola V. de Carvalho et al. has dealt with Docetaxel loaded in copaiba oil-nanostructured lipid carriers as a promising DDS for breast cancer treatment. The manuscript is interesting and well-organized. Moreover, the data presented in the manuscript is quite enough to prove their claims. Based on the critical evaluation, the manuscript is recommended for publication in Molecules. Though, the following changes need to be carried out before the publication.

  1. Why only Docetaxel was used as a capping agent? Why authors did not try other commercially available breast cancer agents? Provide the structure of the molecules for better understanding.
  2. How the present formulation is highly stable under Fe-SEM and TEM environments? Generally, Organic/organic-based formulations are not stable under high energy.
  3. The current formulation is toxic to human normal cells also. How these can be utilized in practical applications?
  4. Some of the important references related to cancer studies need to be cited; ChemMedChem 19 (5), 532 –544; Bioorganic Chemistry 53, 24-36; ChemMedChem doi.org/10.1002/cmdc.202200471. 

Overall, the manuscript needs a major revision. 

Author Response

First of all, we are grateful to the reviewers for their critical reading of the manuscript and important suggestions that have contributed to improve it. We have carefully revised the manuscript, taking into consideration all the reviewers’ comments. Specific answers to each query are supplied below, and changes made to the manuscript are highlighted in the revised version.

Reviewer 1

The manuscript reported by Fabíola V. de Carvalho et al. has dealt with Docetaxel loaded in copaiba oil-nanostructured lipid carriers as a promising DDS for breast cancer treatment. The manuscript is interesting and well-organized. Moreover, the data presented in the manuscript is quite enough to prove their claims. Based on the critical evaluation, the manuscript is recommended for publication in Molecules. Though, the following changes need to be carried out before the publication.

  1. Why only Docetaxel was used as a capping agent? Why authors did not try other commercially available breast cancer agents? Provide the structure of the molecules for better understanding.

Thanks for the comments. You are right, our goal was precisely to investigate the use of natural excipients (with anticancer effects) associated to canonical anti-cancer agents. DTX was chosen because it is the first line drug for breast cancer treatment. In our group we are investigating the encapsulation of other agents (paclitaxel, doxorubicin and cyclophosphamide) in NLC, but for each drug a complete set of DoE has to be conducted, mainly because of miscibility issues, to determine the best composition. So, it is unlike that such different agents would benefit from the specific NLC here reported.

Please notice that the structures of the NLC excipient and docetaxel were included in the revised manuscript (new Figure 1) except for copaiba oil, that contains 14 major compounds.

  1. How the present formulation is highly stable under Fe-SEM and TEM environments? Generally, Organic/organic-based formulations are not stable under high energy.

We appreciate your question. In this work, we used negative staining with uranyl acetate, and then the samples were subjected to a drying process (at 25 °C, for 24 hours) prior to the microscopic analysis, which contributed to reduce the impact of the inherent dehydration of the vacuum chamber. In addition, for TEM, the lowest possible energy / electron beam source (60 kV) was chosen, to guarantee the shortest exposure time and quick acquisition. We have followed the reports of other authors that, using suitable preparation methods combined with effective microscopic setups achieved successful TEM and FE-SEM analyses of lipid nanoparticles (eg doi.org/10.1016/j.micron.2012.07.008; doi.org/10.1208%2Fs12249-009-9197-2; doi.org/10.3844/ajptsp.2008.219.224; doi.org/10.1016/j.ijpharm.2006.08.007). A similar methodology was also employed previously in our group, with good results and sample stability for nanostructured lipid carriers (doi.org/10.1038/s41598-020-76751-6; doi.org/10.1002/jctb.6715) and liposomes (doi.org/10.1016/j.ijpharm.2021.120944).

  1. The current formulation is toxic to human normal cells also. How these can be utilized in practical applications?

In this article only one normal cell line was used: murine fibroblastic NIH-3T3 cells. This lineage is the model cell lineage for cytotoxicity studies, being more used than similar human lineages such as hFF1 cells (doi:10.3390/ph13120463). The results in Fig. 7 show that the optimized formulation is less toxic than commercial DTX to NIH-3T3 cells. Even that NLCCTL showed some toxicity (due to CO) against murine fibroblasts, it is mild compared to that promoted by commercial DTX, that contains detergent micelles (Tween 80) plus 13 % ethanol to solubilize it.

Moreover, cytotoxicity assays are not enough to determine the toxicity of systemic treatments in vivo (doi.org/10.1016/j.ejpb.2016.08.001). The toxicity of nanoparticle is determined by drug bioavailability associated with in vivo oncological models (doi.org/10.1016/j.ejpb.2016.08.001). Studies show that once translated to in vivo assays, the toxicity of these treatments can decrease due to recognition by the reticuloendothelial system (https://pubs.rsc.org/en/content/articlelanding/2022/bm/d2bm00181k /unauth) combined with the accumulation of the nanoparticles in tumors favored by the Enhanced Permeability and Retention (EPR) effect (doi.org/10.3390/jpm11080771).

  1. Some of the important references related to cancer studies need to be cited; ChemMedChem 19 (5), 532 –544; Bioorganic Chemistry 53, 24-36; ChemMedChem doi.org/10.1002/cmdc.202200471. Overall, the manuscript needs a major revision. 

Thank you for the suggested references. We have incorporated some of them in the revised manuscript (line 78). As for the second manuscript, we hope that the corrections introduced in this revised version - in response to each of the reviewers ’points - have sufficiently corrected the problems detected in the original version.

Reviewer 2 Report

The paper outlines the loading of the chemotherapeutic, docetaxel (DTX), into nanostructured lipid carriers (NLCs) for the specific purpose of drug delivery for breast cancer treatment. It reports standard pharmaceutical methods and concepts. The clinical application of DTX remains limited due to its poor aqueous solubility and nonspecific distribution. In addition, DTX acts as a P-gp substrate; consequently, the intracellular DTX concentration is reduced by drug efflux leading to low efficacy and high drug resistance. With DTX being a class IV drug, there are numerous research articles describing strategies to improve its delivery., with recent papers focusing on micro and nanoencapsulation strategies, so the conceit is not new. The novelty is claimed to be the use of copaiba oil for the NLCs, acting as a therapeutic excipient. It was selected for its anticancer and analgesic properties. The paper is generally well written and describes appropriate formulation  development, characterisation and cell studies

It was tested in a murine breast cancer cell line.

Formulation was determined using design of experiments with CO as the liquid lipid and myristyl myristate and Miglyol 812.

122.  Give reference for 20mV being the magnitude of ZP stability of these formulations

149 Span values smaller than unit confirmed..please clarify

178 lay inbetween

192 disappeared

194 Rewrite Moreover, the diffraction patterns of the particles containing or not DTX were similar,

206. Conform statistical analysis used to demonstrated lack of significance.

225 taking almost six times

240, zero order describes a continuous rate of release. The rate of release in the figure does not look zero order

I would question the experimental set-up for release studies. Use of Franz cells is unusual as they are normally for absorption and permeation studies. Why was this membrane used? As the pore size is large, this would not be a barrier. It would be better to run a dialysis-type set up, like the references quoted to support this. How has the fitting of these models led to a better understanding of the mechanism ruling DTX liberation? It has established the kinetics but the mechanism is not discussed,

Release equations should use consistent nomenclature.

286 Since NLCDTX was found more cytotoxic than Synth-NLCDTX formulations, for the three cell lines tested (Figure 6 B,D,F), a closed look at the effect of copaiba oil was advised-rewrite this

309 “But unfortunately the standard deviation of the determined IC50 values were too large and compromised the assessment of CO and DTX additive effects into the NLCDTX” This is due to variability in wither the performance of the formulation or in the cells’ response to them

458. but for MCF cells. Incomplete?

461. Is it synergy or just additive?

Author Response

First of all, we are grateful to the reviewers for their critical reading of the manuscript and important suggestions that have contributed to improve it. We have carefully revised the manuscript, taking into consideration all the reviewers’ comments. Specific answers to each query are supplied below, and changes made to the manuscript are highlighted in the revised version.

 Reviewer 2

The paper outlines the loading of the chemotherapeutic, docetaxel (DTX), into nanostructured lipid carriers (NLCs) for the specific purpose of drug delivery for breast cancer treatment. It reports standard pharmaceutical methods and concepts. The clinical application of DTX remains limited due to its poor aqueous solubility and nonspecific distribution. In addition, DTX acts as a P-gp substrate; consequently, the intracellular DTX concentration is reduced by drug efflux leading to low efficacy and high drug resistance. With DTX being a class IV drug, there are numerous research articles describing strategies to improve its delivery., with recent papers focusing on micro and nanoencapsulation strategies, so the conceit is not new. The novelty is claimed to be the use of copaiba oil for the NLCs, acting as a therapeutic excipient. It was selected for its anticancer and analgesic properties. The paper is generally well written and describes appropriate formulation development, characterization and cell studies it was tested in a murine breast cancer cell line. Formulation was determined using design of experiments with CO as the liquid lipid and myristyl myristate and Miglyol 812.

122 Give reference for 20mV being the magnitude of ZP stability of these formulations

We apologize for the mistake. Please notice that references with that information have been include (line 133) in the revised manuscript.

149 Span values smaller than unit confirmed..please clarify

  1. Rephrased: “The observed Span values smaller than 1 confirmed the homogeneous size distribution of the optimized formulation [55], in agreement with the DLS results”(lines 166-168).

178 lay inbetween

OK, corrected: “Finally, the endothermic peaks determined in the optimized formulations (NLCCTL = 51.53 °C and NLCDTX = 50.21 °C) were found broadened and between those of the NLC major excipients, as an evidence of their (MM, P68) molecular rearrangement inside the lipid nanoparticles (please see lines 195-197).

192 disappeared

Correction made. Thank you for this remark.

194 Rewrite Moreover, the diffraction patterns of the particles containing or not DTX were similar

OK, Rephrased (lines 211-212).

206 Conform statistical analysis used to demonstrate lack of significance.

Correction made. Thank you for this remark.

225 taking almost six times

OK (lines 247-248).

240  zero order describes a continuous rate of release. The rate of release in the figure does not look zero order

The KinetDS 3.0 software (doi:10.14227/DT190112P6) was employed to fit the kinetic curves through several mathematical models. The results of such analyses have been included in the supplementary materials, as Table S3. According to it, highest determination coefficient (R2 = 0.9486) was assigned to zero-order model, that describe the release of an active agent as a function of time at a constant rate (doi:10.1016/b978-0-08-100092-2.00005-9). Zero order described the release profile of low soluble drugs from lipid nanoparticles (doi.org/10.1016/B978-0-08-100092-2.00005-9) and other lipid and polymeric nanoparticles (doi.org/10.1007/s40005-017-0320-1), as cited in the text. As for the experimental set-up, (Franz cells, with donor-acceptor compartments) please consider the answer to the next question.

I would question the experimental set-up for release studies. Use of Franz cells is unusual as they are normally for absorption and permeation studies. Why was this membrane used? As the pore size is large, this would not be a barrier. It would be better to run a dialysis-type set up, like the references quoted to support this. How has the fitting of these models led to a better understanding of the mechanism ruling DTX liberation? It has established the kinetics, but the mechanism is not discussed,

Franz cells are largely used as the analytical devices in drug release analyses of nanocolloids (doi.org/10.1007/s11095-018-2513-3) and both the FDA Guidance for Industry (SUPAC-SS) and USP General Chapter (<1724> Semisolid Drug Products—Product Performance Tests) recommend the vertical diffusion cells to perform in vitro release testing of semisolid formulations (eg creams, gels, lotions and ointments). The Franz-cell device can be applied both for drug release as for permeation studies, the major difference residing in the nature of the used membranes; for drug release studies, synthetic (polycarbonate) or cellulose membranes (dialysis bags) can be used to separate the donor and acceptor compartments. The experiment is really close to a dialysis set up, but the device allows the use of lower volumes (0.2:15 mL donor-acceptor), and easy sample removal/replacement from the acceptor compartment to keep the Sink condition (doi.org/10.1016/j.ijbiomac.2018.12.130; doi: 10.1016/j.msec.2020.]111774; doi: 10.1016/j.biomaterials.2013.02.014). The porous size of the polycarbonate membrane (47 mm diameter, 100 nm molecular exclusion pore, Nucleopore Track-Etch Membrane, Whatman®) employed in the experiments of Fig. 6 was smaller than those of the nanoparticles, to guarantee that only DTX could cross it. Such information has been included in the methodology (line 456).

Finally, a discussion on the mechanism of drug release from the NLC was included at the end this topic (lines 266-272).

Release equations should use consistent nomenclature.

Thank you for this remark. The nomenclature of equations 4 and 5 have been merged (lines 468-471).

286 Since NLCDTX was found more cytotoxic than Synth-NLCDTX formulations, for the three cell lines tested (Figure 6 B,D,F), a closed look at the effect of copaiba oil was advised-rewrite this

OK, corrected (line 317).

309 “But unfortunately the standard deviation of the determined IC50 values were too large and compromised the assessment of CO and DTX additive effects into the NLCDTX” This is due to variability in within the performance of the formulation or in the cells’ response to them?

  1. We first thought that the variability in IC50 values could be related to the cell lineages, since the SD values of MCF-7 cells were always higher and these cells have slower proliferation kinetics than the two other lineages (Sutterland, Hall & Taylor, Cancer Res 43:3998, 1993, PMID: 6871841).

We reported the results obtained after 24 h interval of treatment because in the literature this is the standard time for cell cytotoxicity essays. But even at experiments at longer times of treatment (72 h) the results were not enough to explain the variability (even that SD values decreased a little).

It is interesting that high SD were detected for both (natural and synthetic) NLC, in comparison to free DTX. So, taking into consideration the high % EE of DTX in the formulations, we believe the variability was due to the cells’ response to DTX treatment, mainly the sustained release of docetaxel from the nanoparticles (notice that the variability was not high in formulations without docetaxel – see SD of the IC50 values of Table 3).

458 but for MCF cells. Incomplete?

  1. We rephrased de sentence.

461 Is it synergy or just additive

  1. Thank you very much for the questioning. The Chou-Taladay method for drug combination (doi.org/10.1158/0008-5472.CAN-09-1947) defines a combination index, CI, to differentiate the mechanisms of drug interaction (CI < 1, synergy; CI = 1, additivity; and CI > 1, antagonism). Since at the concentrations (0.3 and 3 mM) the cytotoxic effects of NLCDTX over the 4T1 cell were greater than those of DTX alone (Figure 7 B), the term synergistic is the most suitable, as also pointed out by Yan et al. who investigated the cytotoxic effects of DTX and curcumin on prostate cancer cells (PC-3) (doi.org/10.3109/10717544.2015.1069423). As for the CO effect, the IC50 values decreased when it was encapsulated in the nanoparticles, mainly in the presence of DTX (Table 3). These results are in agreement with reports on natural compounds: when loaded in NLC their effects are improved, possibly due to enhanced bioavailability (doi.org/10.2174/0929867326666190614123835). We have replaced “additive” by synergism throughout the manuscript.

Round 2

Reviewer 1 Report

All the comments were thoroughly revised. The manuscript can be accepted now for publication without any further modifications